# A Molecular Pinball Machine of the Plasma Membrane Regulates Plant Growth—A New Paradigm

**DOI:** 10.3390/cells10081935

**Published:** 2021-07-30

**Authors:** Derek T. A. Lamport, Li Tan, Marcia J. Kieliszewski

**Affiliations:** 1School of Life Sciences, University of Sussex, Falmer, Brighton BN1 9QG, UK; 2Complex Carbohydrate Research Center, University of Georgia, Athens, GA 30602, USA; tan@ccrc.uga.edu; 3Department of Chemistry and Biochemistry, Ohio University, Athens, OH 45701, USA; kielisze@gmail.com

**Keywords:** arabinogalactan protein, proton pump, calcium signalling, auxin, PIN proteins, morphogenesis, Hechtian oscillator

## Abstract

Novel molecular pinball machines of the plasma membrane control cytosolic Ca^2+^ levels that regulate plant metabolism. The essential components involve: 1. an auxin-activated proton pump; 2. arabinogalactan glycoproteins (AGPs); 3. Ca^2+^ channels; 4. auxin-efflux “PIN” proteins. Typical pinball machines release pinballs that trigger various sound and visual effects. However, in plants, “proton pinballs” eject Ca^2+^ bound by paired glucuronic acid residues of numerous glycomodules in periplasmic AGP-Ca^2+^. Freed Ca^2+^ ions flow down the electrostatic gradient through open Ca^2+^ channels into the cytosol, thus activating numerous Ca^2+^-dependent activities. Clearly, cytosolic Ca^2+^ levels depend on the activity of the proton pump, the state of Ca^2+^ channels and the size of the periplasmic AGP-Ca^2+^ capacitor; proton pump activation is a major regulatory focal point tightly controlled by the supply of auxin. Auxin efflux carriers conveniently known as “PIN” proteins (null mutants are pin-shaped) pump auxin from cell to cell. Mechanosensitive Ca^2+^ channels and their activation by reactive oxygen species (ROS) are yet another factor regulating cytosolic Ca^2+^. Cell expansion also triggers proton pump/pinball activity by the mechanotransduction of wall stress via Hechtian adhesion, thus forming a Hechtian oscillator that underlies cycles of wall plasticity and oscillatory growth. Finally, the Ca^2+^ homeostasis of plants depends on cell surface external storage as a source of dynamic Ca^2+^, unlike the internal ER storage source of animals, where the added regulatory complexities ranging from vitamin D to parathormone contrast with the elegant simplicity of plant life. This paper summarizes a sixty-year Odyssey.

## 1. Introduction—Brief Historical Perspective

Sixty years ago [1], the discovery of hydroxyproline (Hyp) firmly bound to the cell wall was the “founder event” for a new field in plant biology. Proteins specific to the cell wall had not previously been considered as components of an otherwise polysaccharide structure, apart from occasional hints in the literature. It is therefore instructive to recall that this discovery originated in D.H. Northcote’s lab adjacent to that of Fred Sanger [2]; his Nobel Prize for protein chemistry in 1958 inspired cell wall protein analyses. Those showed most of the hydroxyproline was localised in the primary cell walls isolated and purified from the first cell suspension cultures generated from cambial explants of sycamore. Thus, a tree reverts to its algal ancestors, provoking Robin Hill’s remark “Wouldn’t it be wonderful to turn an alga into a tree!”. Indeed, the Hill reaction defined the photolysis of water as the ancient source of atmospheric oxygen [3,4]. On the evolutionary timescale, an increased atmospheric oxygen level eventually led to its use as a terminal electron acceptor but also a direct source of the hydroxyproline hydroxyl. The biosynthesis of hydroxyproline, first shown in sycamore cell suspensions, involved the direct incorporation of ^18^O_2_ into the hydroxyproline hydroxyl group [5]. Hence, the first step in understanding the structural and dynamic roles of the Hyp-rich proteins unique to plants was taken. Quite remarkably, mammalian systems have recruited prolyl hydroxylase as an oxygen sensor for hypoxia inducible factor (HIF), which in turn plays a crucial role in foetal development [6].

Fred Sanger’s approach [2] exemplified molecular structure as an initial step towards biological function. That was effective for soluble proteins but not for insoluble hydroxyproline-rich proteins of the cell wall that were only released as fragments [7]. Intensive efforts at solubilisation led to the search for soluble precursors. The candidates included soluble “arabinogalactan polysaccharides” isolated from sycamore cell suspension culture growth media and extracellular polysaccharides from numerous gymnosperms and angiosperm seeds; all contained hydroxyproline but were alanine-rich and lacking tyrosine [8]. The search for a soluble cytoplasmic precursor of wall-bound hydroxyproline (Table 1) yielded an acidic “protein polysaccharide complex” (AGPs), distinguished by its composition and solubility in 10% TCA, which precipitates most proteins [9].

The first analyses of TCA-soluble cytoplasmic protein–polysaccharide complexes purified by preparative isoelectric focusing [9] were subsequently identified as arabinogalactan proteins. Data from [9] show representative species across the plant kingdom included dicots, a gymnosperm and a bryophyte. These were the first AGP analyses notable for their high alanine and low tyrosine, with a high galactose and arabinose content. The amino acid molar ratios were normalized to 30 moles of hydroxyproline compared with the tomato cell wall.

Such proteins, later named arabinogalactan proteins (AGPs), located mainly between the plasma membrane and cell wall, were periplasmic, analogous to Peter Mitchell’s bacterial periplasm [10]. However, they were not precursors to wall-bound proteins, based on their composition and absence of turnover in ^14^C-proline pulse-chase experiments [11].

The identification of the glycopeptide link between protein and polysaccharide depended, again, on a classical Sanger strategy, which involved exploiting the differing stability of various covalent bonds to chemical and enzymic attack. For example, both peptide and glycosidic linkages are acid labile. However, peptide bonds are stable in cold anhydrous HF that cleaves glycosidic linkages. Thus the HF deglycosylation of highly glycosylated proteins [12] enabled the peptide sequencing of difficult proteins such as extensins, AGPs and mucins [13], with subsequent genomic sequencing [14].

Glycosidic linkages are generally stable in mild alkaline environments. That enabled the discovery of the Hyp-O-glycosidic link [15] as Hyp-arabinosides in extensins and Hyp arabinogalactans (Hyp-AGs) in AGPs, respectively. The structural elucidation of Hyp-AGs [16] led to the discovery of their essential role in Ca^2+^ homeostasis, validated by recent direct evidence [17,18,19] and as described in subsequent sections that show how the structure of Hyp-arabinogalactan glycomodules leads to Ca^2+^ homeostasis and plasma membrane Ca^2+^ ion influx that regulates plant growth.

## 2. The Origin of Ion Gradients

Wind and waves generate sea spray aerosols [20] that can act as prebiotic chemical reactors [21]. Prebiotic chemistry generated simple amphiphiles (typically with a hydrophilic headgroup and a hydrophilic tail) that provided a surface hydrophobic membrane that enabled evaporation, the concentration of reactants and ion gradients that included proton gradients. Components of primordial energy transduction systems driven by light may have appeared first in protocells. For example, in purple bacteria such as Halobacterium, bacteriorhodopsin absorbs photons to pump protons across the plasma membrane [22]. Protons initiate both cosmic evolution and biotic evolution; thus, life based on proton gradients is universal.

## 3. Plasma Membrane Dynamics

Over the last billion years or so, a simple lipid membrane evolved from a passive gatekeeper maintaining the “constancy of the interior milieu” to a dynamic control system with a plethora of elaborate import/export microtubule transport systems and kinesin/dynein motors that ferry vesicle cargo to and from the plasma membrane.

Ultimately, three critical ions, H^+^, Ca^2+^ and auxin (indole acetic acid), regulate plant growth. Of these, protons are preeminent players in energy transduction/ATP generation but also maintain the plasma membrane potential of −120 to −160 mV [23]. This creates the electrostatic gradient essential for the influx of cations, particularly the K^+^ osmolyte and the universal signalling ion Ca^2+^, which affects every aspect of a cell’s life and death. Ca^2+^, the most tightly regulated ion within all membrane-bound organisms, binds to thousands of proteins to effect changes in localization, association and function. Hundreds of cellular proteins have been adapted to bind Ca^2+^ over a million-fold range of affinities (nM to mM), in some cases, simply to buffer or lower Ca^2+^ levels and, in others, to trigger cellular processes. The local nature of Ca^2+^ signalling is intimately tied to this large range of affinities [24].

Until quite recently, the mechanism of auxin-induced Ca^2+^ signalling in plants was unclear [25]. Now, however, the subtlety of Ca^2+^ homeostasis and its dynamic storage is apparent: paired glucuronic acid sidechains of AGP Hyp-AG glycomodules bind Ca^2+^ specifically on the outer surface of the plasma membrane and thus connect membrane components involved in Ca^2+^ signalling with auxin-regulated cell expansion; auxin-activated ATPase proton pumps then dissociate AGP-Ca^2+^ supplying the Ca^2+^ channels that regulate cytosolic Ca^2+^. The pinball hypothesis conveniently summarises this scenario of proton “pinballs” ejecting bound Ca^2+^ ions from AGP-Ca^2+^ as discussed in the following sections.

## 4. Proton Pumps

The plasma membrane regulates all aspects of plant growth by an array of mechanisms that include specific receptors, transporters, channels and ion pumps, chiefly proton pumps [26,27]. The Arabidopsis genome encodes eleven proton pumps [23]. That significant redundancy underlies their essential role in maintaining membrane potential and generating the low cell wall pH required by the “acid growth hypothesis”. However, we re-interpret the “acid growth hypothesis”, involving novel roles for the F_1_F_o_ ATPase proton pump in Ca^2+^ homeostasis and auxin transport. This new paradigm, unique to plants, identifies the proton pump as the focal point of many activators and regulators (Figure 1).

The ATPase proton pump generates proton “pinballs” that release Ca^2+^ ions from AGP glycomodules, which supply cytosolic Ca^2+^. Turgor drives cell expansion and activates the pump by the mechanotransduction of wall stress via Hechtian adhesion between the plasma membrane and the cell wall; this creates a Hechtian oscillator that regulates cycles of cell wall plasticity and oscillatory growth. Auxin is the other major proton pump activator transported by auxin-efflux “PIN” proteins. ATPase proton pump regulation is central to plant growth and development [28]. The figure depicts numerous additional inputs. Green arrows indicate upregulation, and red represents downregulation, with exceptions where the mechanism remains unknown. Dotted blue arrows indicate PIN proteins that are directly incorporated into the plasma membrane as integral components. Numerous features underlie the reductionist simplicity shown in the above figure: high auxin levels generally enhance cell extension [29]; auxin-driven morphogenetic patterns depend on unidirectional fluxes [30]; the evolution of auxin signalling and PIN proteins [31]; auxin biosynthesis occurs in meristems [32]; auxin activates the plasma membrane H+-ATPase via phosphorylation [28]. Negative regulation by abscisic acid decreases steady-state levels of phosphorylated H^+^-ATPase, possibly by promoting dephosphorylation via a protein phosphatase [33], and in turn suppresses hypocotyl elongation in Arabidopsis [33]; abscisic acid stress signalling evolved in algal progenitors [34]. Blue light: the blue light photoreceptor pigment phototropin increases cytosolic Ca^2+^ [35]. Brassinosteroids: increase cytosolic Ca^2+^ via increased auxin levels [36]. Cytokinins enhance cell division by unknown mechanisms. Ethylene upregulates auxin biosynthesis in the Arabidopsis root apex and inhibits root cell expansion [37]; thus, anthranilate synthase mutants yield ethylene-insensitive root growth phenotypes. Ethylene specifically inhibits the most rapid growth phase of expanding cells—normally the root hair initiation zone—but ethylene moves it much closer to the tip. Auxin and ethylene act synergistically to control root elongation, root hair formation, lateral root formation and hypocotyl elongation. Ethylene modulates root elongation through altering auxin transport [38]. This figure is a revised version of Figure 2 in [39], animation at https://youtu.be/zABg7LiBk88 (accessed on 30 June 2021)

ATPase H^+^ pumps have two components, F_1_, the ATPase catalytic site, and F_0_, a proton pore; they operate in two modes, “forward” as in mitochondria, where a proton gradient drives the synthesis of ATP, and in “reverse” at the plasma membrane, where ATP hydrolysis drives proton extrusion. In plant cells, proton extrusion plays three major roles: Firstly, the creation of a large membrane potential [~−120 to −160 mV] enhances cation uptake and anion extrusion through the respective channels. Secondly, the proton pump generates protons; arguably, these eject Ca^2+^ from the plasma membrane with AGP-Ca^2+^ glycomodules as the immediate source of cytosolic Ca^2+^. Thirdly, auxin transport involves the efflux of anionic auxin into the wall at low pH where it is protonated and thus becomes neutral. That allows it to diffuse through the plasma membrane into the cytosol of an adjacent cell at a neutral pH, where it re-ionises for efflux and the polar transport of auxin against the concentration gradient [40,41].

Here, we connect an auxin-activated proton pump with AGPs and Ca^2+^ signalling. This reinterprets the “acid growth” hypothesis [42]. The hitherto unsuspected role of proton pumps in Ca^2+^ homeostasis is a new paradigm relevant throughout the Plant Kingdom from tropisms to phyllotaxis [25].

## 5. Indirect Evidence for the Role of AGPs in Ca^2+^ Homeostasis

Michael Jermyn (1975) demonstrated the ubiquity of AGPs (first known as beta lectins) in an impressive range of seed plants [8], suggesting a possible role in signalling per se. Over several decades, a signalling role for AGPs was based intuitively on the complexity of their polysaccharide components and, most significantly, their cell surface location as shown by the remarkable series of anti-AGP “JIM” monoclonal antibodies, developed at the John Innes Institute [43], that recognize specific AGP carbohydrate epitopes. Thus, recent reviews [44,45] cover numerous papers that demonstrate specific AGP distribution, particularly in metabolically active tissues involved in virtually all aspects of plant growth from Bryophytes onwards [46]. AGPs mark critical cell fate transitions [47] in both sporophyte and gametophyte generations. Thus, judging from extensive cytochemical studies, seed germination, vegetative growth, flowering, fertilization [48], embryogenesis [49,50,51], phyllotaxis [52] and fruit ripening [53] all involve AGPs. However structural elucidation of Hyp glycosubstituents was lacking. That limited progress until detailed high-field NMR analyses of purified Hyp-oligosaccharides [54] showed an underlying simple repetitive consensus motif of 15 sugar residues consistent with earlier analyses of AGP carbohydrates. Most significantly, the consensus 15-sugar glycomodule has a short β-1,3-linked galactan backbone with two short mobile sidechains terminated by glucuronic acid that together bind Ca^2+^ stoichiometrically [55]. This pH-dependent Ca^2+^-binding enables peripheral cell surface AGPs to play a role in Ca^2+^ signalling that differs fundamentally from the endogenous Ca^2+^ storage of animal cells. Numerous previous observations of AGP localization and chemical properties throughout the literature are consistent with the new paradigm that unifies AGPs and Ca^2+^ homeostasis with the regulation of plant growth.

An early indication of a connection between AGPs and Ca^2+^ appeared in 1991; the wound response of *Acacia senegal* and its secretory product, gum Arabic, consists of polysaccharides and glycoproteins related to AGPs [56]. Significantly, gum Arabic contains glucuronic acid (~10%) and binds approximately ~1% by weight of Ca^2+^. In the same year [57], the analysis of isolated plasma membranes revealed a bound AGP content of ~10% *w*/*w*. These AGPs were hydroxyproline-rich, with a significant glucuronic acid content (~10%). However, the crucial connection between Hyp-glycosubstituents and the pH-dependent Ca^2+^ binding property of AGPs with Ca^2+^ homeostasis only appeared quite recently when a molecular model depicted paired glucuronic acid residues of a Hyp-glycomodule that bound Ca^2+^ in a molecular dynamics simulation, which was then confirmed through an in vitro assay [55].

## 6. Direct Evidence for AGP Regulation of Ca^2+^ Homeostasis—A New Paradigm

The molecular pinball machine is a visual metaphor where “PIN” proteins trigger the machine by supplying auxin that activates the proton pump with the release of proton “pinballs” that initiate Ca^2+^ influx. This falsifiable hypothesis makes several testable predictions: AGP glucuronic acid is essential for growth.AGP glucuronic acid enables AGP-Ca^2+^ binding.AGP-Ca^2+^ binding is the source of cytosolic Ca^2+^.Specific Ca^2+^ channels facilitate its influx.Auxin triggers a rapid increase in cytosolic Ca^2+^.Ca^2+^ waves are essential for root growth.AGP adaptation to salt stress also involves upregulation of their genes.

Numerous other physiological predictions include tropisms and phyllotaxis [25]. In vitro pH-dependent Ca^2+^ binding by AGPs [55] implied in vivo significance by predicting that Ca^2+^ binding by paired glucuronic acid residues of Hyp-AG glycomodules is the major source of dynamic cytosolic Ca^2+^. When theory predicts, experiments decide. Therefore, plants should exhibit defective Ca^2+^ homeostasis if they lack AGP–glucuronic acid residues (i.e., if AGP glycomodules were not glucuronidated). However, generating AGPs that lack glucuronic acid was a considerable experimental challenge fraught by the redundancy of multiple glucuronosyl transferases (GTGlcA) in the Arabidopsis GT14 family. Both Dupree’s [17] and Showalter’s [18] groups resolved that problem brilliantly, by generating triple GTGlcA knockouts. Dupree’s group generated *glcat14a/b/e triple mutant plants*, while the complementary approach by Showalter’s group generated a *glcat14a glcat14b glcat14c triple glcat* CRISPR-Cas9 mutant line. Those benchmark papers describe how mutants generated AGPs that lacked glucuronic acid; their compelling evidence for the essential role of AGPs in Ca^2+^ homeostasis is as follows:

### 6.1. AGP Glucuronic Acid Is Essential for Growth

Plants have a huge investment in AGPs represented by about 85 genes in the Arabidopsis genome. Two groups [17,18] reported that triple glucuronosyl transferase knockouts decreased AGP glucuronidation; the corresponding decrease in AGP-Ca^2+^ binding was associated with profound developmental defects: trichome branching, suppressed growth phenotypes, decreased inflorescent stem length and a drastic decrease in progeny. Physiological defects included sterility, abnormal Ca^2+^ transients, the attenuation of Ca^2+^ wave propagation and decreased responses to ROS activation of Ca^2+^ channels in roots.

### 6.2. AGP Glucuronic Acid Enables AGP-Ca^2+^ Binding

AGPs isolated from triple glucuronosyl transferase knockouts contained significantly less glucuronidation, with a corresponding decrease in Ca^2+^ binding by AGPs in vitro.

### 6.3. AGP-Ca^2+^ Binding Is a Major Source of Cytosolic Ca^2+^

Earlier work [55] showed enough periplasmic AGP for a significant increase in cytosolic Ca^2+^ and further corroborated by [17,18].

### 6.4. Auxin Increases Cytosolic Ca^2+^

A report that cellular Ca^2+^ signals generate defined pH signatures in plants [58] presents a conundrum of how to discriminate between stimulus and response. In this instance, it was resolved by the observation [59] that auxin elicits a Ca^2+^ signal (aequorin blue fluorescence) “immediately” after the addition of an auxin activating the proton pump, i.e., within seconds, possibly involving the enhanced diffusion of protons along the surface of the membrane, hence their delayed escape from the membrane surface into the adjacent bulk phase [60].

### 6.5. Ca^2+^ ATPase Recycles Cytosolic Ca^2+^

While the pinball machine and Hechtian oscillator emphasise influx, Ca^2+^ homeostasis demands a finely balanced efflux evidenced by the fifteen Ca^2+^-ATPase pumps located in both the ER and plasma membrane of Arabidopsis [61].

### 6.6. Ca^2+^ Waves Are Essential for Root Growth

The Dupree group also examined the dynamic aspects of Ca^2+^ homeostasis in roots. Using the fluorescent reporter R-GECO1, they observed organised cytosolic Ca^2+^ wave propagation at the inner and outer zones of wild-type roots. However, wave propagation was notably disorganized in *glcat14a/b/e* mutant roots. They extended these observations to the H_2_O_2_ activation of Ca^2+^ channels during growth that had been reported in a seminal paper [62]. The Dupree group confirmed that H_2_O_2_ rapidly increased cytosolic Ca^2+^, particularly at the growing tip of wild-type roots. However, this increase was much less in *glcat14a/b/e* mutant roots (Figure 11 in ref [62]), which is consistent with the pinball hypothesis. However, the conclusion that H_2_O_2_-activated Ca^2+^ channels are the major regulator of cell expansion [62] needs to be re-evaluated; decreased H_2_O_2_-induced Ca^2+^ influx of mutant roots shows that Ca^2+^ channels are necessary but not sufficient for complete cytosolic Ca^2+^ influx. Thus, we conclude that Ca^2+^ channels are essential components of the molecular pinball machine that needs both AGP-Ca^2+^ and activated Ca^2+^ channels for maximal activity. This also applies to other aspects of plant growth including the tip growth of root hairs and pollen tubes.

### 6.7. AGPs Respond to Salt Stress

An increasing sodium ion concentration competes with bound AGP-Ca^2+^ and so may account for salt-sensitive crop plants such as rice. On the other hand, salt-resistant plants typified by salt-marsh flora and marine seagrass raise the question of mechanism. AGPs are involved in two well-documented instances: firstly, AGP upregulation is a direct response to increased salt levels in plant cell suspension cultures [63]; secondly, some higher plants, exemplified by *Zostera marina,* have returned to a marine habitat [64]. Zostera architects of the seagrass meadow “blue carbon” ecosystem provide an exquisite test of the pinball hypothesis. How do plants growing in such high-salt environments cope with the serious competition of Na^+^ ions with Ca^2+^ ions? Some plants such as Zostera have adapted by increasing their glucuronic acid content [65], hence increasing their ability to bind Ca^2+^. With recent advances in genetic engineering in mind, the possibility of re-engineering AGPs to enhance salt-sensitive crop plants is apparent.

## 7. The Quest for Key Regulators of Plant Growth

Oscillatory growth implies that a biochemical oscillator regulates the process, with protons and Ca^2+^ ions as key players [66]. It also includes AGPs, auxin and auxin efflux “PIN” proteins. However, the idea of a Ca^2+^ signature that encodes specific developmental cues has become quite attractive. Some, such as [67], suggest that specificity *“involves one messenger with many translators*” and is encoded in Ca^2+^-signalling systems by Ca^2+^-binding proteins with different Ca^2+^ affinities. This may enable cells to decode an initial Ca^2+^ signal to yield graded or nuanced responses. They also noted [67] the apparent evolutionary loss of Ca^2+^-influx systems in plants but suggested that “Ca^2+^-influx mechanisms and components were still to be discovered”. The molecular pinball machine fills the gap as an essential component of Ca^2+^ signalling and homeostasis; in quiescent cells, auxin and Ca^2+^ trigger pinball activity. Thus, “PIN” proteins supply auxin that activates the proton pump to promote expansion growth. However, the Ca^2+^ signal depends not only on AGP-Ca^2+^ but also on Ca^2+^ channels activated by reactive oxygen species (ROS) [62] and stress transduction from the cell wall to mechanosensitive ion channels [68,69,70] of the plasma membrane. The transmission of the stress vector most likely involves an arabinogalactan protein APAP1 [71] covalently attached to wall pectin and anchored to the plasma membrane by a C-terminal GPI lipid. That provides a physical basis for Hechtian adhesion and mechanotransduction. Hence, the turgor pressure powers a Hechtian oscillator and components of the pinball machine that regulate cell expansion. This raises the great question of cell wall plasticity that remains unanswered after sixty years since James Bonner and his colleagues [72] began the quest for “Haftpunkte” load-bearing bonds that, when broken, allow cell wall loosening. Although expansins [73] soften the wall, the precise biochemical mechanisms remain unknown, even after thirty years of trying to “round up the usual suspects” such as hydrogen bonds [74]. Underlying assumptions must always be questioned. Specific load-bearing covalent bonds in the wall may not exist in a wall stretched by viscoplastic flow or “creep”. If we view the primary cell wall as analogous to a plastic, then a plasticizer, such as expansin, may control wall rheology [75]. However, the primary cell wall, with its multiple interactions, is a complex multicomponent structure, rather like a molecular chess board, which accounts for the slow progress in solving the problem. Pectin is the major component of the primary cell wall, with its direct involvement likely, although its subtle chemistry suggests unknown mechanisms. Our current hypothesis invokes the sycamore cell suspension culture [76], which is of historical interest [77], not only as the first primary cell wall to be isolated [1], but also as the initial source of AGPs as polysaccharides [78] and hydroxyproline-rich glycoproteins [9]. Sycamore cell cultures characterised as finely divided pipettable suspensions [77] were always accompanied by soluble macromolecules including AGPs released into the culture medium during growth [9,78]. AGPs released by the phospholipase cleavage of their GPI anchor may act as wall plasticizers by disrupting the alignment of linear pectin macromolecules. Another possibility highlights the versatility of AGPs that depends on their high affinity for Ca^2+^ [55]. Thus, pectin methyl esterase deesterifies pectin, resulting in free carboxyls crosslinked by Ca^2+^ and a more rigid wall. However, AGPs in muro with a higher affinity for Ca^2+^ than pectic carboxyls effectively scavenge pectate Ca^2+^. This results in negatively charged carboxyl ions such that pectin macromolecules then repel each other [75] and thus loosen the wall. The small 22 kDa diffusible AGP peptides [79] may also be effective scavengers of Ca^2+^. This may account for the effect of low pH increasing wall plasticity by dissociating calcium pectate Ca^2+^ ions that are then captured by AGPs. That resolves the acid growth paradox.

## Figures and Tables

**Figure 1 cells-10-01935-f001:**
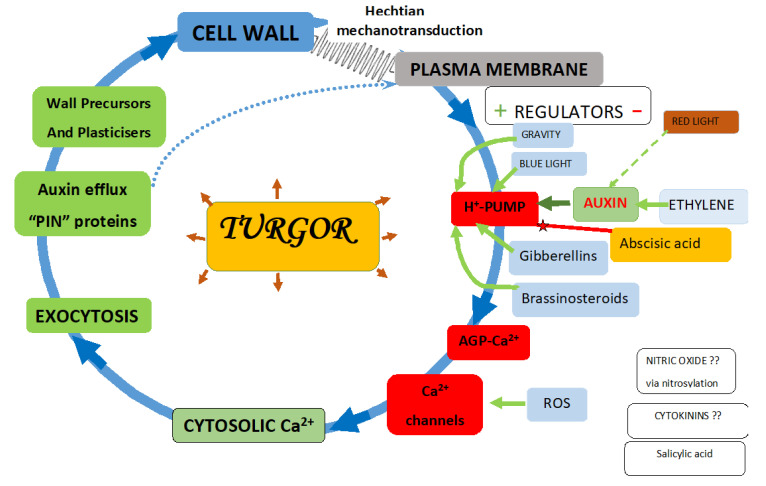
The Hechtian growth oscillator incorporates a molecular pinball machine.

**Table 1 cells-10-01935-t001:** Composition of the TCA-soluble cytoplasmic fraction.

Amino Acid	Sycamore	Tomato	Sphaerocarpos	Ginkgo	Tomato Cell Wall
**Hyp**	**29**	**31**	**30**	**28**	**30**
Pro	6	nd	nd	1	8
Asp	11	5	15	25	8
Thr	14	12	17	16	6
**Ser**	**19**	**19**	**20**	**27**	**15**
Glu	8	6	10	20	9
Gly	9	9	10	14	8
**Ala**	**20**	**26**	**27**	**28**	**7**
Val	6	7	15	9	8
Cys	6	4	4	5	0
Met	0	1	3	2	1
Ile	4	2	1	4	5
Leu	6	4	8	10	9
**Tyr**	**1**	**0.5**	**1**	**2**	**3**
Phe	3	0.5	2	3	3
**Lys**	7	4	3	6	11
His	1	0.5	1	2	2
Arg	2	1	1	2	4
Galactose	[++++]	740	[++++]	[++++]	150
Arabinose	[++++]	540	[++++]	[++++]	165

++++: these sugars were highly abundant.

## Data Availability

Not applicable.

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
