# Peer review of "A Molecular Pinball Machine of the Plasma Membrane Regulates Plant Growth—A New Paradigm"

_cells, 2021, doi:10.3390/cells10081935_

Round 1

Reviewer 1 Report

In this review, the authors have discussed the role of plant AGP in the regulation of Ca2+ signalling with particular attention to auxin response. I have found the review of the interest and somehow provocative. I think there is still a need for more confirmation to fully support the authors’ model, but I think this review would help scientists design future experiments.

I do have a few comments that could be useful to improve the manuscript.

L21. “Mechanosensitive Ca2+ channels and their activation by reactive oxygen species (ROS) are yet another factor regulating cytosolic Ca2+…”

I would better explain this sentence. What is the evidence the authors are referring to?

L79. “alkali” is repeated twice.

L141. I think that the authors can also consider the work by Barbez et al., 2017 (Apoplastic pH regulation in A. thaliana roots Barbez et al.,  Proceedings of the National Academy of Sciences Jun 2017, 114 (24) E4884-E4893)

L148. I do not get the point of reporting the following sentence: “Blue light: the blue light photoreceptor pigment phototropin increases cytosolic Ca2+ [35].

L150. “Ethylene upregulates auxin biosynthesis in the Arabidopsis root apex and inhibits root cell expansion [37]…”. Actually, it might be also true the opposite. Auxin induces the expression of ACS genes in root cells (Tsuchisaka and Theologis, 2004 (Unique and overlapping expression patterns among the Arabidopsis 1 amino cyclopropane 1 carboxylate synthase gene family members. Plant Physiol. 136: 2982 3000).

L164. “Secondly, the proton pump generates protons that eject Ca2+ from the plasma membrane AGP-Ca2+ glycol modules as the immediate source of cytosolic Ca2+”.

I think this sentence is quite strong. If what the authors have written has been demonstrated there is the need to cite the work they are referring to.

L172. “The hitherto unsuspected role of proton pumps in Ca2+ homeostasis is a new paradigm relevant throughout the Plant Kingdom from tropisms to phyllotaxis.”

To add another piece in the puzzle is it interesting to note that Ca2+-ATPase to export Ca2+ from the cytosol to the apoplast besides using ATP acts by a net Ca2+/H+ exchange lowering the energetic requirement for the export (Rasi-Caldogno et al., 1987 The Ca-transport ATPase of plant plasma membrane catalyzes a nH/Ca exchange. Plant Physiol.  83: 994–1000).

L199. “Acacia senegal” instead of “Acacia Senegal”.

L244. “AGP-Ca2+ binding is the source of cytosolic Ca2+.”

I would tone down a little bit this sentence. There are indeed other sources of cytosolic Ca2+ in the plant cells (e.g. vacuole and ER lumen), and I would not exclude that in the apoplast besides a role played by AGPs also pectins could be involved.

L252-254. “….after addition of an auxin activating the proton pump, i.e. within seconds possibly involving enhanced diffusion of protons along the surface of the membrane hence their delayed escape from the membrane surface into the adjacent bulk phase [60]”.

I am sorry but I did not really get the meaning of this part. Could the authors better explain this point?

I think the authors might also take into consideration the work of Dindas et al., 2018 (AUX1-mediated root hair auxin influx governs SCFTIR1/AFB-type Ca2+ signalling. Nat Commun 9, 1174) where it is shown that in root cells IAA can affect the cytosolic pH due to the AUX1 transport activity.

L258. “…located in both ER and plasma membrane of Arabidopsis…”. Besides the ER Ca2+-ATPases are present in the tonoplast and Golgi Apparatus. Some recent reviews were published (Bonza and De Michelis, 2011, The plant Ca2+-ATPase repertoire: Biochemical features and physiological functions. Plant Biol (Stuttg) 13: 421–430; Resentini et al., 2021 The signatures of organellar calcium. Plant Physiol. 26:kiab189. doi: 10.1093/plphys/kiab189; He et al., 2021 Transport, functions, and interaction of calcium and manganese in plant organellar compartments. Plant Physiol 889 doi.org/10.1093/plphys/kiab122).

L264. “…that had been reported in a seminal paper [62].”. Before Foreman et al., (2003), Pei et al., 2000 (Calcium channels activated by hydrogen peroxide mediate abscisic acid signalling in guard cells. 1067 Nature 406: 731–734) reported the discovery of H2O2-activated Ca2+ currents in the plasma membrane of guard cells.

L327. Remove “by”.

Reviewer 2 Report

This review revisits the theory of acidic growth in plants by introducing the role of arabinogalactan glycoproteins (AGPs). It relates the historical stages of discovery of the role of AGPs. It is a quality review which will be very useful for the physiology of the plant. However, I think these authors shouldn't just repost the 2018 figure of their previous article, and should update it again. I suggest a MINOR REVISION before publication.

Reviewer 3 Report

Dear Authors,

the detailed comments are in the PDF file. Here, I put the main comments:

1/ line 100 - This place is too big shortcut, suggesting that clathrin coated vesicles are part of the exocytic pathway; meanwhile, clathrin is part of the endocytic pathway; please explain this shortcut or add appropriate literature.

2/ line 133; the scheme/figure 1 should be explained in more detail, so that even for a reader with little knowledge of the subject it is understandable; please explain where the plasma membrane is, what is meant by a dotted arrow, a dashed arrow (from red light), a red line with an asterisk (from abscisic acid to H+-pump).

Since the diagram/figure 1 is almost identical to the one already published, I understand that the Authors have obtained permission from the publisher; if not, I do not pay attention to this fact.

I also suggest improving the quality of the figure 1; for example, the arrows on the blue line are misaligned with the line; correct the arrow shape from "gravity" to H+ - pump.
